# An Experimental Study of Damage Detection on Typical Joints of Jackets Platform Based on Electro-Mechanical Impedance Technique

**DOI:** 10.3390/ma14237168

**Published:** 2021-11-25

**Authors:** Liaqat Ali, Sikandar Khan, Naveed Iqbal, Salem Bashmal, Hamad Hameed, Yong Bai

**Affiliations:** 1College of Civil Engineering and Architecture, Zhejiang University, Hangzhou 310058, China; drali169@zju.edu.cn (L.A.); hamadhameed32@gmail.com (H.H.); baiyong@zju.edu.cn (Y.B.); 2Department of Mechanical Engineering, King Fahd University of Petroleum and Minerals, Dhahran 31261, Saudi Arabia; bashmal@kfupm.edu.sa; 3Department of Electrical Engineering, King Fahd University of Petroleum and Minerals, Dhahran 31261, Saudi Arabia; 4Center of Energy and Geo Processing, King Fahd University of Petroleum and Minerals, Dhahran 31261, Saudi Arabia; 5Interdisciplinary Research Center for Intelligent Manufacturing and Robotics, King Fahd University of Petroleum and Minerals, Dhahran 31261, Saudi Arabia

**Keywords:** electromechanical impedance, piezoelectric transducers, damage detection, non-destructive test, Q345B steel materials, welded steel joints

## Abstract

Many methods have been used in the past two decades to detect crack damage in steel joints of the offshore structures, but the electromechanical impedance (EMI) method is a comparatively recent non-destructive method that can be used for quality monitoring of the weld in structural steel joints. The EMI method ensures the direct assessment, analysis and particularly the recognition of structural dynamics by acquiring its EM admittance signatures. This research paper first briefly introduces the theoretical background of the EMI method, followed by carrying out the experimental work in which damage in the form of a crack is simulated by using an impedance analyser at different distances. The EMI technique is used to identify the existence of damage in the welded steel joints of offshore steel jacket structures, and Q345B steel was chosen as the material for test in the present study. Sub-millimetre cracks were found in four typical welded steel joints on the jacket platform under circulating loads, and root average variance was used to assess the extent of the crack damage.

## 1. Introduction

The electromechanical impedance (EMI) method is a recent non-destructive evaluation (NDE) method that utilizes a single piezoelectric material to behave both as actuator and sensor. The piezoelectric material produces an electric charge when exposed to mechanical stress. The electrical impedance of piezoelectric material is directly associated with the mechanical properties of the host structure. The variations of dynamic properties in the host structure caused by the presence of damage also influence the measured impedance plots, which are used in the process of damage assessment [1]. The piezoelectric material is widely used due to its light weight and availability in various sizes and shapes [2,3]. Among the mentioned methods in the literature for structural health monitoring, the EMI technique is considered the best promising approach due to its high sensitivity to local incipient damage [3,4]. This technique acquires variations in local dynamics due to incipient structural damage at the proper level. These variations are very complicated and very small and are not easily detectable with the ordinary low-frequency vibrations techniques. The EMI method can be implemented very easily to identify the damage in the host structure [4].

In the literature, various applications of the EMI technique have been discussed, such as damage detection of the steel composite materials and concrete structures using a piezoelectric transducer (PZT) patch connected to the host structure. The need for structural damage monitoring has increased more in the past few years due to the excessive use of composite materials as construction materials [5,6,7,8]. Sun et al. [9] have used the EMI technique for detection of the integrity of the laboratory-sized truss. Pearis et al. [10] proposed a new version of the EMI technique using the FFT analyser with improved qualities. This new version of the EMI technique was further expanded by Xu et al. [11] and Panigarhi et al. [12]. Tseng and Naidu [13] performed an experimental study for the damage detection using the PZT transducers on aluminium specimens, and they concluded that the PZT transducers have a remarkably effective area for detection of small initiatory damages. The EMI technique was also used by Giurgiutiu [14] for the damage detection of an aging aerospace structure, and it was concluded that the EMI technique has a wide prospect in detection of damages in the aerospace field. Several studies in the literature have discussed a wide range of successful applications of damage detection for various types of structures for the impedance-based method [15,16,17,18,19,20,21,22,23].

With development of marine resource exploration, many large-scale marine structures, for instance, oil rigs, undersea tunnels, and cross-sea bridges, must be constructed. Unlike onshore structures, offshore structures are often affected not only by workloads but also by other loads from storms, waves, tides, and corrosion of seawater. These offshore facilities need under-water welding during the manufacturing phase and also for repair [24,25]. High-Strength-Low-Alloy (HSLA) steel is commonly used in building construction, offshore construction, and automotive industries [26,27,28]. Using HSLA steel will excessively reduce the welding costs [29,30,31,32]. Part of the structure that works under water may need repair work. Amongst the three most important methods in water welding, i.e., wet method, dry chamber hyperbaric weld method, and coffer dam system method, the majority use the wet welding method, where the electric arc, as well as joints, are in direct interaction through water [33,34,35].

The wet welding can be done by covered electrodes and flux connector cables. As a welding atmosphere, water causes many problems in obtaining the quality of the joint [36,37,38,39,40,41]. The main significant problem in the undersea welding of martensitic structures is the Heat-Affected-Zone (HAZ) [42,43]. The construction of the welding joint has the characteristics of low plasticity and high rigidity. Another major problem is the occurrence of diffuse hydrogen content in welded metals [44]. The hydrogen comes from water vapour, as well as from electrode coverings. Compared to the H5 or H10 levels of land welding, the hydrogen content in the joins under wet welding circumstances is at the 50–80 mL/100 g placed in the metal range [45,46,47,48,49,50]. Di et al. [51] performed a study in which the sheets of Q345B steel were welded, and the microstructure of the joints was analysed. The results show that the presence of the internal defects in the welded Q345B steel joints depends on the stir–pin rotation speed, weld speed, and cooling intensity during welding. Mihajlo et al. [52] performed both numerical and experimental studies in order to understand the fracture behaviour of welded joints with multiple defects. The results suggest that the weakness of the welded joints are due to the misalignment and incomplete root penetration and also due to the unseen internal defects.

Chalioris et al. [53] utilized PZT transducers in various settings to diagnose and demonstrate flexural damages caused by concrete cracking and steel yielding. Two large-scale flexural beams were used that were subjected to monotonically increased loading until failure. Small-sized PZT patches were bonded by epoxy on the steel surface of the tension reinforcing bars and also on the external concrete face of the beams. The damage in the steel was diagnosed from the difference of the response signals from the PZT transducers for the baseline healthy state and for the state when the loading caused the damage. PZT transducers were used for the real time structural health monitoring of reinforced concrete members and for the earthquake damage detection in seismic-prone regions. The comparison of the response signals acquired from the bonded piezoelectric patches for the healthy and the damaged states showed the location and intensity of the damage in the steel bars inside the concrete [54]. The transportation of carbon dioxide during the process of carbon dioxide capture, storage and utilization is carried out by means of pipelines that are normally exposed to a variety of environmental conditions. In order to safely transport the carbon dioxide to the injection point, these pipelines should be prevented from the various types of failures. The fatigue failure may be one of the possible causes of the pipeline’s failure and should be monitored continuously during the process of carbon dioxide capture, storage and utilization [55,56,57,58,59,60]. The offshore pipelines are continuously exposed to the waves in the sea that is a continuous dynamic load and can be a potential cause of the pipeline failure. The underwater structures can failed by a variety of failure mechanisms that include fatigue failure, corrosion failure, internal sheath damage etc. The enhanced Risk Based Inspection (RBI) methodology can be used to maintain the integrity of steel pipeline that are used in offshore applications [61,62,63,64,65,66].

This paper is focused on the experimental study about the electromechanical impedance (EMI) technique used for the damage crack detection in four types of welded steel joints for offshore jacket platform structure. In all the specimens, the sub-millimetre size cracks were successfully detected and estimated using the root mean square deviation (RMSD) method.

## 2. Theory of Electro-Mechanical Impedance Techniques

Liang et al. [67] studied the theoretical background of the EMI techniques for the first time. A single PZT Transducer of 10 mm^2^ size was used to acquire the response of a host structure that was excited above 20 kHz. The PZT was connected to the host structure using an adhesive, and 1 V of low voltage was utilized to excite the structure. The 1-D model given in Equation (1) relates the reverse impedance of the electrical PZT with the mechanical impedance of the structure. Any possible changes in the properties of the structure can be identified by continuously monitoring the changes in the electrical impedance of the PZT transducer attached to the structure. The terms ω, α, ε33 T, $d_{3x}^{2}$, $\gamma_{\mathrm{xx}}^{E}$, and δ represent the input frequency, geometric constant, dielectric constant, coupling constant, loss tangent, and Young’s modulus, respectively. Due to the temperature sensitivity of the dielectric constant ε33 T, the imaginary part of the impedance will be affected, and therefore, only the real part of the impedance will be used for the EMI method [9,68].
(1)γ(ω)=iωα(ε33 T(1−iδ)−Zs(ω)Zs(ω)+Za(ω)d3x2γxxE)

Various other studies in the literature have also worked on the theoretical aspects of the EMI method while considering the system as 1-D [69,70,71,72,73]. Zagrai and Giurgiutiu [74] considered a 2-D model for a circular 2-D structure and validated it against the experimental outcomes. Their model considered axial and flexural vibrations of a target structure by taking into consideration the sensor dynamics. The suggested model is given in Equation (2). The term *k*^2^*_p_* is the planar coupling, v is Poisson’s ratio, *J*_0_ and *J*_1_ are the Bessel functions of the first kind of order zero and one, respectively.
(2)$$Z\left(\omega \right)\left\{i\omega C\left(1-k_{p}^{2}\right)\times \left[1+\frac{k_{p}^{2}}{1-k_{p}^{2}}\frac{\left(1+v\right)J_{1}\left(\varphi_{a}\right)}{\varphi_{a}J_{0}\left(\varphi \right)-\left(1-v\right)J_{1}(\varphi_{a}-x\left(\omega \right)\left(1+v\right)J_{1}\left(\varphi_{a}\right)}\right]\right\}$$

A 2-D impedance model was proposed by Bhalla et al. [75], in which the shear lag effect, produced by the adhesive bond layer between the PZT transducer and the host structure, was considered. Figure 1 demonstrates the 2-D impedance model with the shear lag effect. The model given in Equation (3) was compared to work of Bhalla and Soh [76], which clearly explained the phenomenon of shear lag. In Equation (3), the term *T* is the complex tangent ratio that is theoretically equal to tan (*k*_l_)/*k*_l_, where *k* is the wave number, and $Z_{a,eff}$ and $Z_{s,eff}$ are the actual impedance of the PZT and the host structure, respectively [76].
(3)$$\bar{Y}=G+Bj=4\omega j\frac{l^{2}}{h}\left[\varepsilon_{33}^{\bar{T}}-\frac{2d_{31}^{2}{\bar{Y}}^{E}}{1-v}+\frac{2d_{31}^{2}{\bar{Y}}^{E}}{1-v}\left(\frac{Z_{a,eff}}{Z_{s,eff,eq}+Z_{a,eff}}\right)\bar{T}\right]$$

The EMI technique is normally applied by utilizing impedance analysers such as Agilent 4194A. Due to the huge costs of these impedance analysers, low-cost ways are proposed in various studies in the literature. Peairs et al. [10] present a costs efficient method for employing the EMI technique using the FFT analyser with a simple circuit. In this method, the input voltage and the current are used to calculate the impedance (Z). Figure 2 shows the above-mentioned simple circuit.

The key operational principle of the EMI technique is the monitoring of the variations in the mechanical impedance of the host structure from the PZT element’s electrical impedance. The EMI technique utilizes high frequencies in the range of 20–400 kHz [50]. As shown in Equation (4), Girugiutiu [77] used the Root Mean Square Deviation (RMSD) method to quantify the intensity of the structural changes.
(4)RMSD=(∑k=1N[(Re(Zk)j−Re(Zk)i]∑k=1N[Re((Zk)i))12 
where, $\left(Z_{k}\right)$*i* and $\left(Z_{k}\right)j$ are the reference impedance of the PZT at the *k*th measurement point [76].

## 3. Experiment Setup

Four types of specimens were used in laboratory measurements: The first one is welded steel plates, manufactured at Marine Engineering Experimental Institute at Zhejiang University, Hangzhou, China, prepared by welding the two steel plates of same size in horizontal direction with an electrode having similar composition to the metal being welded, as shown in Figure 3a. The second one is T-type welded steel plates, manufactured at Marine Engineering Experimental Institute at Zhejiang University, Hangzhou, China, prepared by welding the two metallic plates of different sizes in such a way that the first plate is placed in a vertical direction and second smaller plate is placed in a horizontal direction, as shown in Figure 3b. The third specimen is the combination of the plate and a pipe of the T-type welded metallic joint, manufactured at Marine Engineering Experimental Institute at Zhejiang University, Hangzhou, China, prepared by welding the two metallic plates and pipe of different sizes in two different directions, such that the longer plate is placed in vertical direction while the smaller pipe is placed in horizontal direction, as shown in Figure 3c. The fourth specimen used in the testing is a combination of two steel pipes of different sizes, manufactured at Marine Engineering Experimental Institute at Zhejiang University, Hangzhou, China, prepared by welding the two steel pipes in a T-type joint in such a way that the longer pipe is placed in a horizontal direction, while the smaller pipe is placed in a vertical direction, as shown in Figure 3d. Q345B is a low alloy structure steel, with good mechanical properties, low temperature performance, plasticity, and conditional weldability. Q345B is mainly used in low-voltage vessels, fuel tanks, vehicles, cranes, mining machinery, power stations, mechanical components, construction structures, offshore platform structures, and general metal parts. The mechanical properties are given in Table 1. Q345B steel plate is always hot-rolled and can be used in areas above −40 degrees centigrade temperature. In order to estimate the weldability of the base metal Q345B, the carbon equivalent (CEQ) was calculated for the base metal using the following equation.
CEQ = C + Mn/6 + Ni/15 + Cu/15 + Cr/5 + Mo/5 + V/5(5)

The chemical composition values needed in Equation (5) are given in Table 2. The CEQ value shows that the Q345B steel has conditional weldability with CEQ value of 0.49% and can be used in the above-mentioned various applications. Based on the weldability analysis of Q345 steel, the welding parameters and electrode types were selected, as given in Table 3. The material used in all of the specimens is steel Q345B, and the properties are given in Table 1. The chemical composition of Q345B steel is given in Table 2.

CO_2_ gas shielding is used to weld the specimens. Before connecting components, the surface of steel parts is cleaned with an electric grinder, manufactured at Marine Engineering Experimental Institute at Zhejiang University, Hangzhou, China, for corrosion, oil stains, oxide scales, and other substances harmful for welding. Steel plates must be pre-heated properly before the welding to prevent hardened structures and cold cracks. The preheating of the area around the weld joint or the entire part to a specified temperature before welding will help in reducing the cooling rate of the weld and drives out moisture, which in turn helps prevent hydrogen build-up and cracking. The specimens were heated in an oven, manufactured at Marine Engineering Experimental Institute at Zhejiang University, Hangzhou, China, within a temperature range of 100 to 150 degrees centigrade based on the base metal chemistry as specified in Equation (5). The heat input value of welding in the water environment is determined using Equation (6), excluding the welding efficiency factor.
Heat Input (Q) = U∗I/V(6)

The values of the parameters needed in Equation (6) are given in Table 1, Table 2 and Table 3. Welding parameters are provided in ranges. The magnitude of the selection of welding parameter from a given range depends on the welding quality requirements. Macroscopic image of the welded joint is shown in Figure 4. From the side view of the welded joint, in Figure 4, the thickness of the heat-affected zone is shown. It can be seen that the welding is efficiently done with an optimum thickness of the heat-affected zone. The main focus of the current study is to detect the activation of cracks in already welded specimens. The variation in the impedance of the impedance tester is an indication of the damage in the welded specimens. The maximum value of the impedance corresponds to the occurrence of the damage in the welded specimens. Growing damage causes a shift of most resonance peaks, which is the result of increasing dynamic changes of the host structure when the crack is introduced. Cyclic loading, acting on the host structure, continuously changes its dynamical properties, and therefore, the EMI results from each damage case would be different from other cases for a specific host structure during the entire loading condition.

Table 3 also provides information about the thermal input (kJ/mm). If defects, such as pores and slag inclusions, are present after the welding, the defective parts must be cleaned with an electric portable grinder and repaired with manual arc welding.

### 3.1. Specimens and Load

Each specimen was loaded with a different load, and they showed different reactions to the loading conditions. Specimen B03 was loaded with a load of 180 kN magnitude, which showed good resistance in the case of tension, but it cannot withstand the compressive load. Specimens I13 and H09 were loaded by a load of 150 kN magnitude to detect the crack. It should be noticed that the load condition of T04 is quite different from other situations. It was loaded by a load of 100 kN magnitude for a fixed period of cyclic loading to detect the crack. Different load conditions of the four specimens are shown in Table 4.

### 3.2. Experimental Arrangement

An experimental process was performed in the Marine Engineering Experimental Institute at Zhejiang University. The experimental setup consists of an impedance tester (WK6500B), manufactured by Wayne Kerr Electronics, UK, a PC programmed with VEE Pro software, developed by Keysight Technologies, USA, having the ability to handle MATLAB details for signal preparation and information analysis, and an integrated piezoelectric lead zirconate titanate structural system, as shown in Figure 5.

The WK6500B impedance tester is incorporated to get the impedance signature from the PZTs. The applied quality management system of the WK6500B impedance analyser is in line with BS EN ISO 9001, which can measure the impedance from 20 Hz up to 120 MHz and is capable of performing fully automated high-speed testing. The impedance analyser is connected to the PC by a cable (LAN JR45-type) to get the measured results. The PC used in the experiment was programmed with VEE Pro software. PZTs were pasted on each specimen by using some special power epoxy gel to have good contact with the surface. A pair of PZT transducers, manufactured by Piezo Direct, CA, USA, were hooked up to the impedance analyser in a two-terminal pair configuration. The Material Test System (MTS) load frame machine is connected through a computer to control the applied load and to measure the strain produced in the specimen during testing. Each specimen is gripped by the Materials Test Systems (MTS) load frame machine, manufactured by MTS Systems, MN, USA, and monotonically loaded in tension. The exact gripped location at both sides of the specimen was marked by markers, and the length of the specimen was measured between the jaws of the load frame in order to inspect the enlargement due to the cyclic tensile loading process.

The experimental setup is shown in Figure 6. This setup is located in the Marine Engineering Experimental Institute at Zhejiang University. Once the PZT are attached to the specimens, the specimens are loaded in the load frame machine. All the specimens were checked twice by using the impedance analyser for any possible error before applying the load. The experimental procedure in the current study is designed based on the guidance provided in the ASTM E466-15 standard force-controlled fatigue tests, ASTM E647-15e1 standard for measurement of fatigue crack growth rates, E739-10(2015) standard for statistical analysis of linear or linearized stress-life (S-N) and strain-life (ε-N) fatigue data, and E1049-85(2017) standard for cycle counting in fatigue analysis. The cyclic load from the load frame is continuously applied on the specimen until the crack is detected.

Each specimen bounded with PZT transducers was loaded in the MTS load frame machine as shown in Figure 6. Before applying any tensile load, the specimens were checked twice by an impedance analyser to get the impedance signature for an intact case for making sure that the specimen is good for testing, and there is no difference between both results. Tension load is applied to levels of 1.5% strain. The data of the undamaged case were saved in the PC memory as a baseline signature of the host structure as in the “as-received” condition. To detect the crack, a penetration test is applied after having some change in length of specimen, and the specimen is analysed using a digital microscopic camera. The results were displayed by VEE Pro software on the computer screen. If the cracks are not detected by a digital microscopic camera, the MTS load frame machine is continued again for cyclic tension load until the crack has been detected. The test is terminated after the detection of a crack. DPT-8 dye penetrate inspection material is used to perform the penetration test to create a more visible surface of specimen for detection by digital microscopic camera, manufactured by Dino-Lite, CA, USA.

The location of the damage affects the sensitivity of PZT. If the damage location is closer, then the PZT is more sensitive. In the case of specimen B03, PZT is pasted on some distances away from the welded zone, due to cyclic tensile loading, because it cannot remain attached during the whole experiment, while, in other cases of T-type specimens (I11, H09, and T04), PZT is pasted near to the welded zone for achieving better impedance results. Cracks were detected and measured in each specimen by using a microscopic camera. The variations in the dynamic properties of the host structures, caused by the occurrence of damage, influence the recorded impedance plots.

## 4. Results and Discussion

Digital macro-tests have shown that all the welded joints have many defects, which are typical for underwater conditions [46,81,82]. The most common detected are undercuts and cracks. Garg et al. [83] concluded that increasing the cooling rate can lead to an increase in the number of cuts in the electro-weld joints. High cooling rates under wet welding conditions result in bottom cuts in all the specimens. Digital macro-tests of Controlled thermal severity (CTS) specimens showed differences compared to pad welding observations made in previous surveys [84]. Cold cracks in welded joints are caused by different thermal conditions. These differences are related to the different shapes of the specimen and presence of notches in the CTS sample [85,86].

In the current study, the recorded admittance signatures of EMI obtained at each PZT show the variation caused by the presence of damage and provide good information for damage localization. The damage detected by the digital microscopic camera is shown in Figure 7 and Figure 8. Arrow signs shown in Figure 7 and Figure 8 depict the crack location in different cases of the EMI-based damage assessment testing. All the specimens were successfully inspected for the sub-millimetre cracks.

The developed EMI-based technique has the efficient capability of detecting the presence of cracks in the welded specimens. The variation in the impedance of the impedance tester is an indication of the damage in the welded specimens. The variation in the impedance for the four specimens is shown in Figure 9. In Figure 9, the variation in the impedance is given for the intact case as well as for the other cases with different cyclic loading time periods (T). The maximum value of the impedance in Figure 9 corresponds to the occurrence of damage in the welded specimens.

The destruction can largely be noticed by the modifications in the holding evaluations. It was expected that certain PZT sensors far away from the welded zone and crack location would register less modification in the electromechanical holding signature as compared to the sensors adjacent to the welded zone of the crack location. Figure 9 shows a clear identification of variation in the impedance by comparing the electromechanical impedance responses to the intact case for the host structure. Growing damage causes a shift of most resonance peaks, which is the result of increasing dynamic changes of the host structure when the crack is introduced. Cyclic loading, acting on the host structure, continuously changes its dynamical properties, and therefore, the EMI results from each damage case would be different from other cases for a specific host structure during the entire loading condition.

As shown in Figure 9, most of the variations in the impedance occur between the frequency ranges of 140–170 kHz. It can be seen from Figure 9 that the magnitude of the impedance varies excessively for the three types of cyclic periods with respect to the intact case. The maximum variation at a certain cyclic period shows the occurrence of damage in that specific specimen.

It can be realized by looking at the results of experiments that the damage has caused a shift in the EMI peak signature. Usually, when the damage appears, it reduces the system’s rigidity, and consequently, it develops the transpose of the entrance top that shows a sign of damage, which confirms the feasibility of the EMI type destruction analysis method. According to Figure 9, the cyclic period (T) of loading for specimens represents different result cases of E/M impedance signatures.

To estimate the validity of EMI-based damage detection methodology, the Root Mean Square Deviation (RMSD) damage detection metric from Equation (4) is used to evaluate the accuracy of experimental results. According to the RMSD metric, the receptivity and susceptibility of actual and fictional sections of the PZT’s EM signatures for crack analysis have been logically observed, and overall outcomes uphold the description. The values of the RMSD metric of the impedance signature were calculated for each damage case. The damage was modelled as a vertical notch for each damage case. The RMSD histograms are given in Figure 10 and Figure 11, which demonstrates the variation of RMSD with an increase in the destruction severity for specimens B03, I13, H09, and T04. From the graphs can be seen that RMSD’s results for specimens B03, I13, H09, and T04 show an explicit inclination of damage severity towards the lower damage cases.

For all the analysed destruction cases, an important plane nexus was viewed among the destruction indicator worth and mark volume and size, except for specimen T04 as it may be caused by the special fixed form and loading conditions. Based on the trend of the bar, it can be observed that the RMSD destruction indication may smartly display the modifications of the EM impedance entrance signs among the intact and discovered destruction cases. The largest RMSD value of damage cases represents very effectively the occurrence of damage. In other words, the RMSD analysis provides significant data for destruction detection. The usability of the developed systemic fitness surveillance mechanism is very common for detecting breaks in engineering systems in various environmental conditions.

Impedance-based methods might be applied for existing aging structures, as well as for future new structures. Besides that, productive individuals are not needed to forecast the electromechanical impedance signs, as they need not be very efficient in knowing the mechanical concepts of structural properties and in the statistical estimation of the RMSD metric. The sensitivity of the electromechanical impedance technique is very commensurable as compared to ultrasonic techniques. Despite that, various ultrasonic approaches failed to detect fractures perpendicular to the wave propulsion surface [16,87]. To detect damages, the EM impedance-based techniques are very efficient in significant cost savings and safety enhancement through the wide area of implementation. In the upcoming period, the EMI method will be significantly connected with extra methods, especially wireless techniques for more productive structural health monitoring.

## 5. Summary

This paper is focused on the experimental study of the electromechanical impedance (EMI) technique used for damage detection in four types of welded steel joints for offshore jacket platform structures. In all the specimens, the sub-millimetre size cracks were successfully detected and estimated using the RMSD method. The outcomes of the current study are summarized below:The admittance signals from PZTs cause the reflection of cracks inside welded steel joints. When the damage appears in the welded zone of the specimen, it influences the recorded impedance plots and presents the variations of dynamic properties in the host structures that help in locating the initiated cracks.The maximum value of the impedance corresponds to occurrence of the damage in the welded specimens. Load acting on the host structure continuously changes its dynamical properties, and therefore, the EMI results from each damage case would be different from other cases for a specific host structure during the entire loading condition.The value of the RMSD smartly displays the modifications of the EM impedance entrance signs among the intact and discovered destruction cases. The largest RMSD value of damage cases represents very effectively the occurrence of damage. The RMSD analysis provides significant data for destruction detection.The EMI technique has the strength of detecting cracks in the welded joints in initial phases which can help in avoiding catastrophic failures of offshore structures.The efficiency of the EMI technique, based on the root means square deviation method, expresses a good capability of the proposed methodology for detecting cracks in welded metallic structures.Based on the crack detection results in the current study, the EMI technique is highly efficient and reliable and can be used in a variety of engineering applications.

## 6. Future Work

Although a lot of research has been carried out related to the EMI techniques, there are still a lot of problems related to their implementation in various applications. Some of these problems are locating and identifying the damage type, the imperfect sensing variety, the requirement to choose the right frequency range, and so on. The artificial neural network (ANN) is a possible solution to the above-mentioned problems because it has given promising outcomes for various similar applications. Along with the use of ANN for measuring the structural damage, it can also be used for a variety of other applications, such as selecting an appropriate frequency range, temperature recompense, forecasting concrete strength, etc. Advancements in computer technologies and the continuous efforts from the scientific community will bring the EMI technique one step closer to commercialization. The current study can further be extended to cover non-destructive testing in various underwater applications.

## Figures and Tables

**Figure 1 materials-14-07168-f001:**
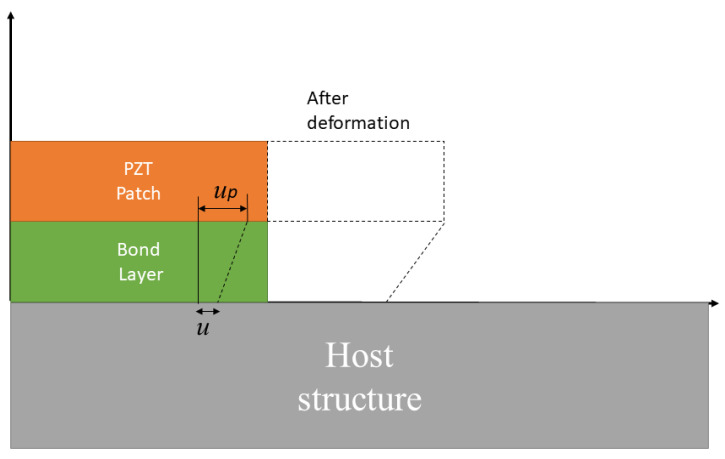
PZT Transducer deformation and bond layer.

**Figure 2 materials-14-07168-f002:**
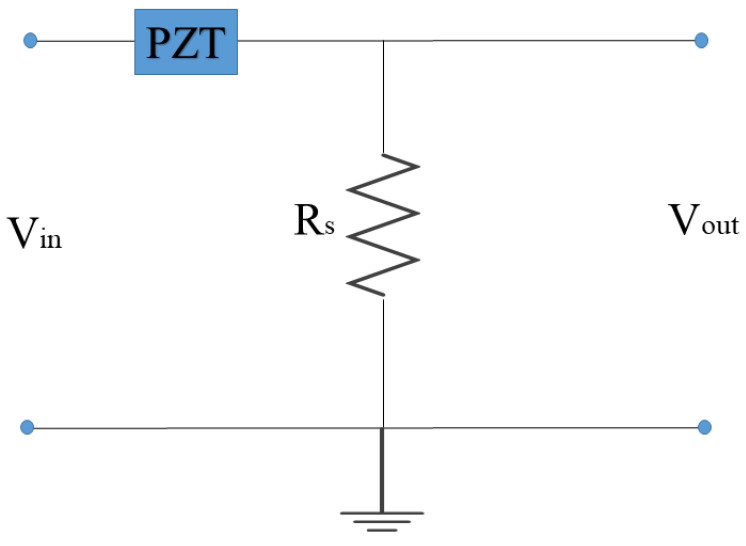
Circuit for calculating the PZT impedance.

**Figure 3 materials-14-07168-f003:**
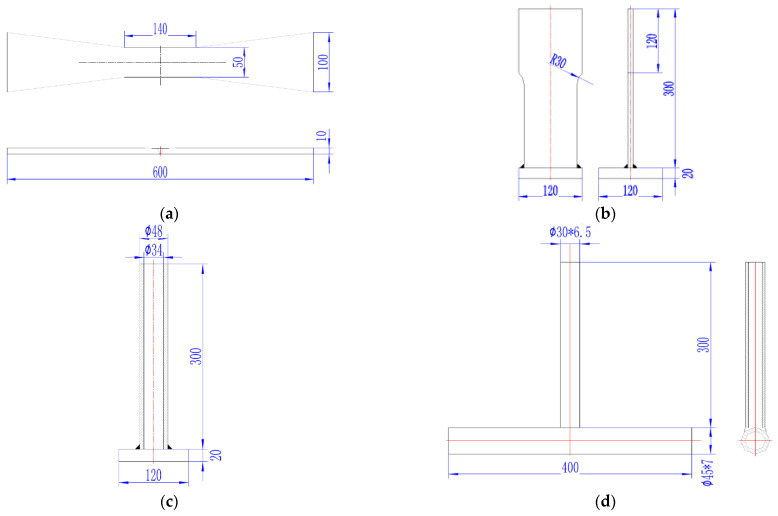
Geometric structure of the specimen used in laboratory measurements (in mm) (**a**) Welded steel plates (B03), (**b**) T-type welded steel plates (I13), (**c**) combination of plate and pipe (H09), and (**d**) combination of two steel pipes (T04).

**Figure 4 materials-14-07168-f004:**
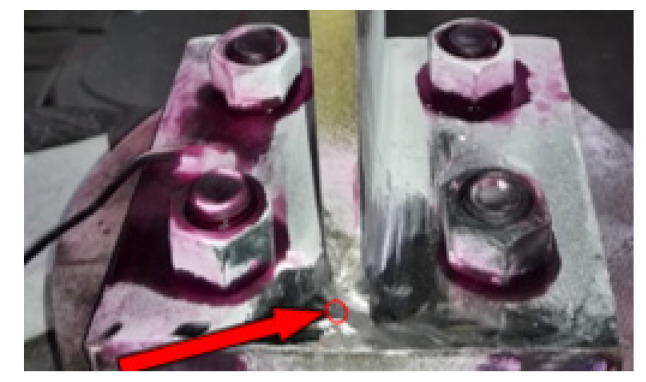
Welded joints with exposed heat affected zone.

**Figure 5 materials-14-07168-f005:**
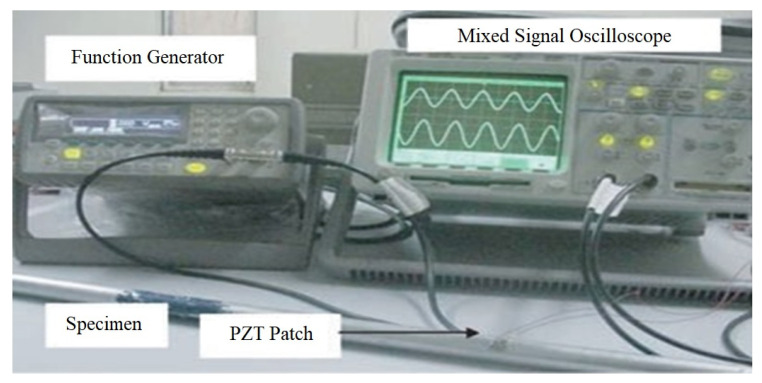
Practical setup for impedance type destruction analysis method.

**Figure 6 materials-14-07168-f006:**
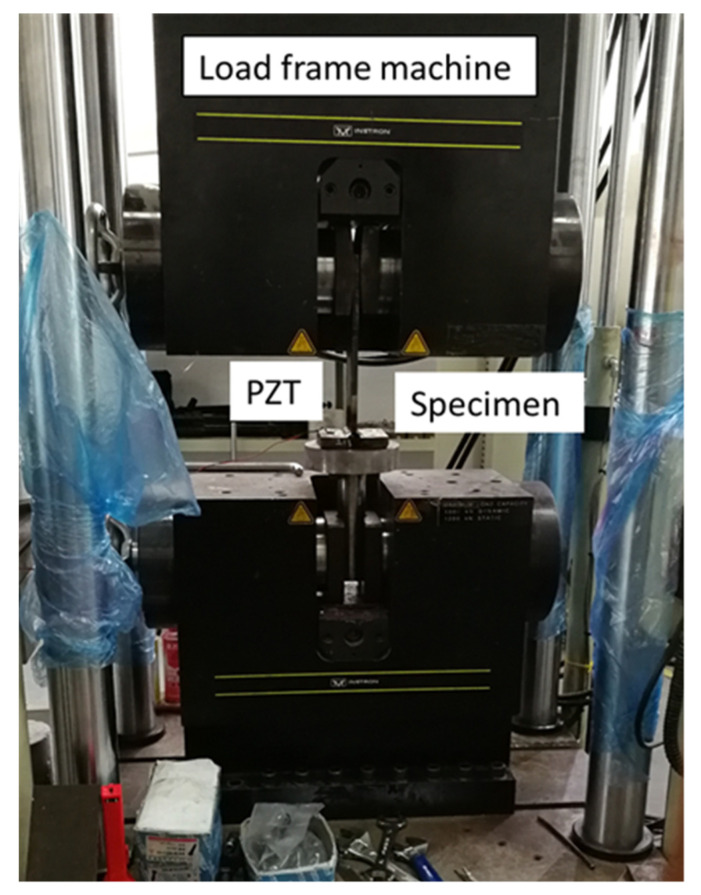
Experimental arrangement.

**Figure 7 materials-14-07168-f007:**
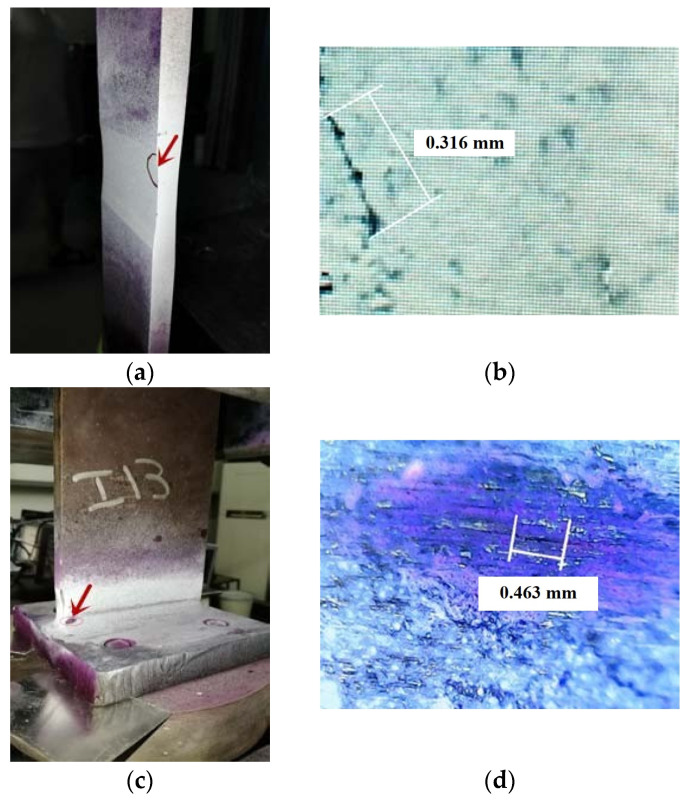
Crack locations in the specimens: (**a**) welded steel plates (B03), (**b**) crack detected in the welded steel plates (B03), (**c**) T-type welded steel plates (113), and (**d**) crack detected in the T-type welded steel plates.

**Figure 8 materials-14-07168-f008:**
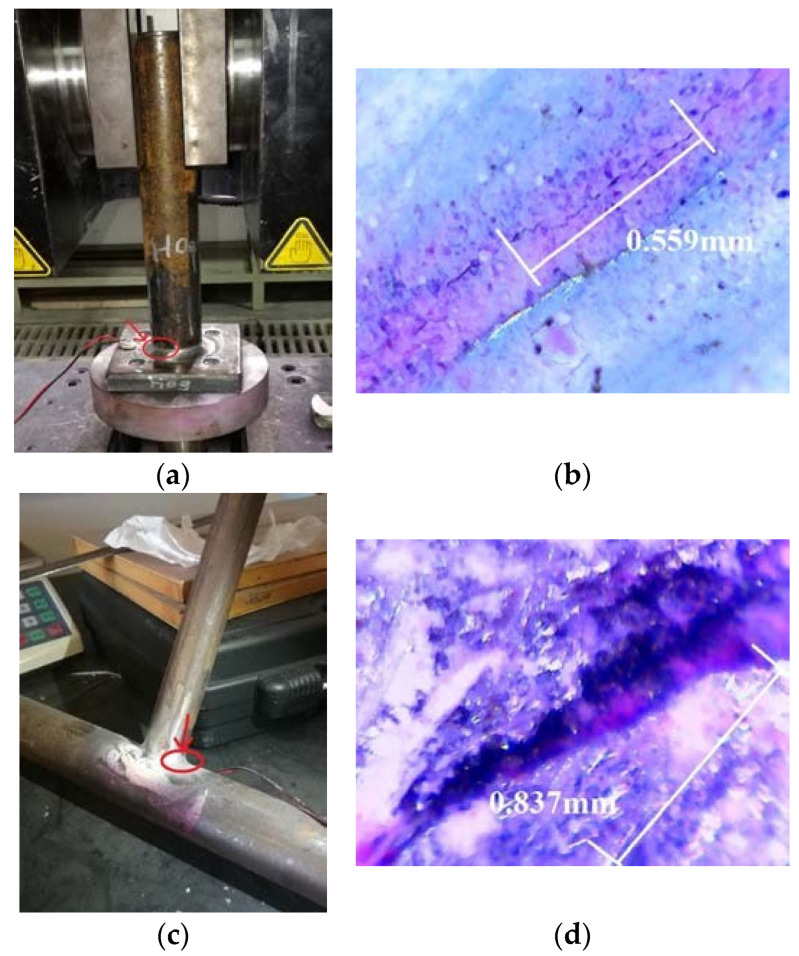
Crack locations in the specimens: (**a**) combination of the plate and a pipe (H09), (**b**) crack detected in the combination of the plate and a pipe (H09), (**c**) combination of two steel pipes (T04) and (**d**) crack detected in the combination of the two steel pipes (T04).

**Figure 9 materials-14-07168-f009:**
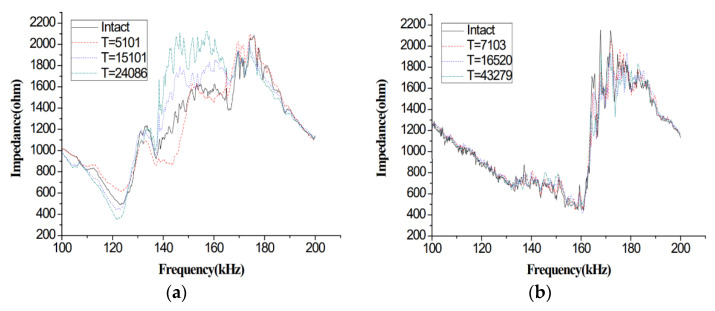
Difference in impedance for the four types of specimens: (**a**) welded steel plates (B03), (**b**) T-type welded steel plates (I13), (**c**) combination of plate and pipe (H09), and (**d**) combination of two steel pipes (T04).

**Figure 10 materials-14-07168-f010:**
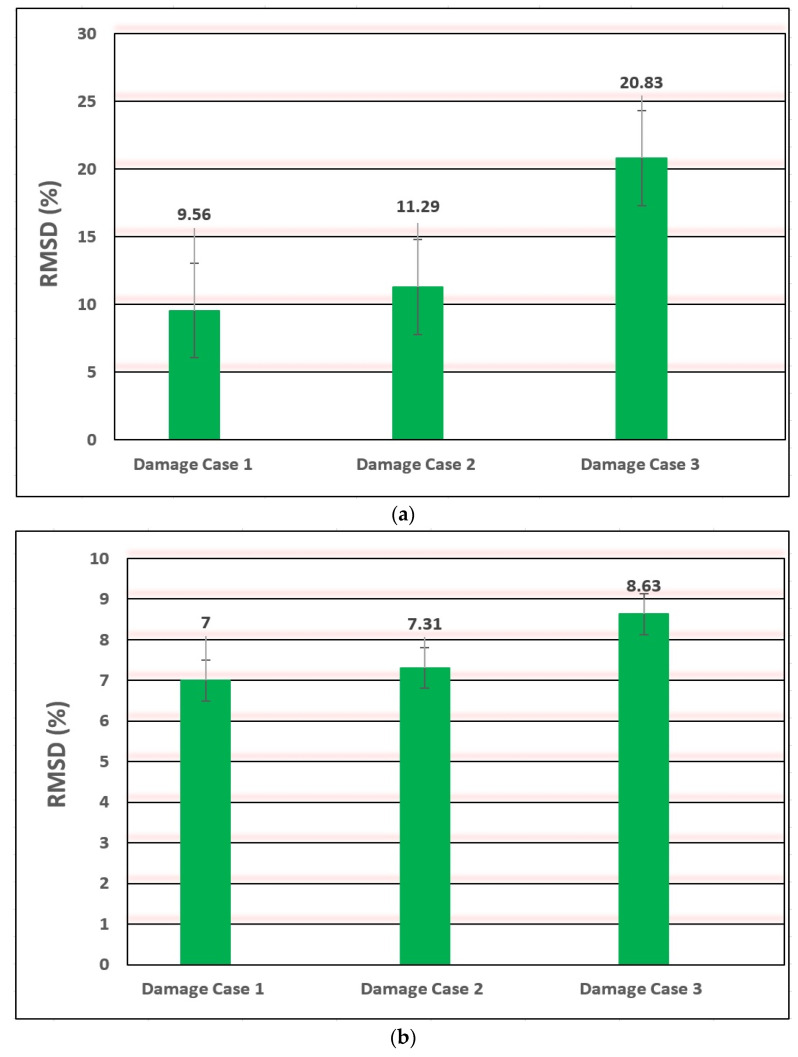
RMSD for damage cases of specimens (**a**) welded steel plates (B03) and (**b**) T-type welded steel plates (I13).

**Figure 11 materials-14-07168-f011:**
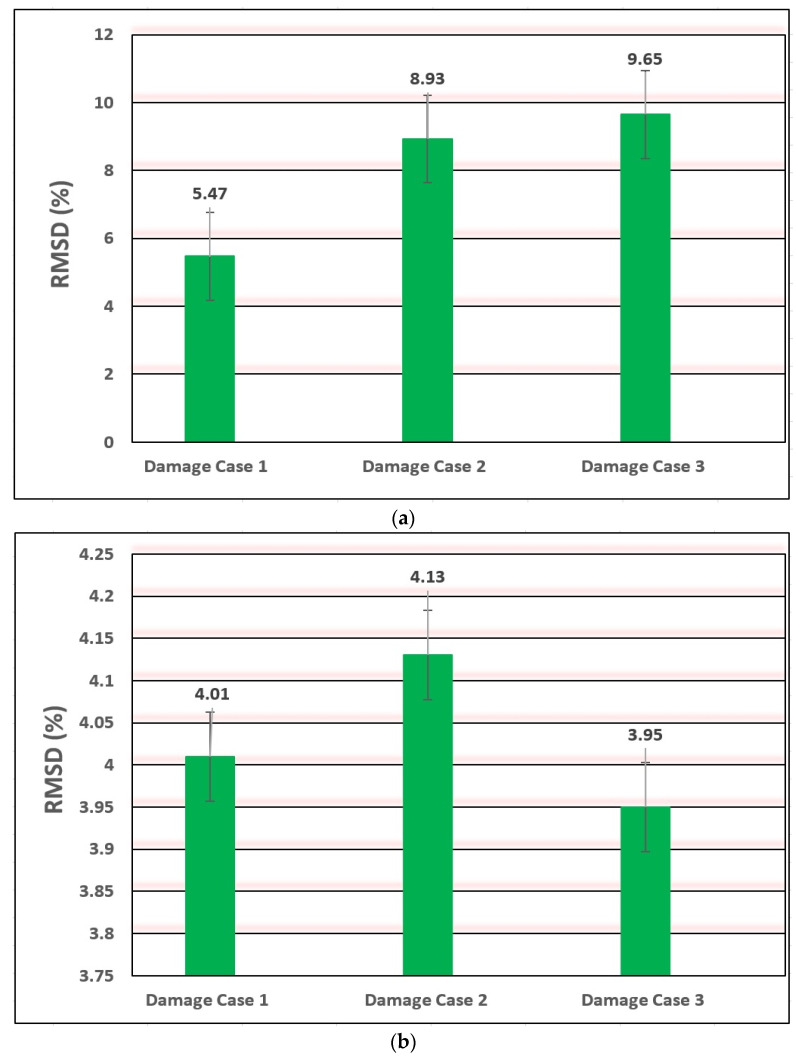
RMSD for damage cases of specimens (a) combination of plate and pipe (H09) and (b) combination of two steel pipes (T04).

**Table 1 materials-14-07168-t001:** Material properties of specimens [78,79,80].

Properties	Value
Tensile Strength	490–675 (MPa)
Yield Strength	≥345 (MPa)
Elongation after Fracture	≥21%

**Table 2 materials-14-07168-t002:** Chemical composition of Q345B steel [78,79,80].

Elements	Contents
C≤	0.2
Mn	1.0–1.6
Si≤	0.035
S≤	0.035
Al≥	0.02–0.15
Nb	0.015–0.06
V	0.02–0.15

**Table 3 materials-14-07168-t003:** Welding parameters.

Specimen	Welding Current [A] 50–100	Arc Voltage [V] 18–21	Welding Speed [mm/s] 5–20	Heat Input [kJ/mm]
B03	90	20.0	6.5	0.28
	75	19.0	5.0	0.27
I13	85	19.5	5.35	0.31
	94	20.5	6.5	0.30
H09	98	18.5	5.0	0.37
	88	21.0	6.18	0.30
T04	95	19.0	6.3	0.35
	82	18.5	5.1	0.31

**Table 4 materials-14-07168-t004:** Load specifications of the specimens.

Specimen	Load (kN)	Note
B03	180	Repeated tension without compression in the case of buckling
I13	150	Push–pull fatigue
H09	150	Push–pull fatigue
T04	100	Push–pull fatigue with fixed period

## Data Availability

The data are available by contacting the corresponding authors.

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
