# Peer review of "An Experimental Study of Damage Detection on Typical Joints of Jackets Platform Based on Electro-Mechanical Impedance Technique"

_materials, 2021, doi:10.3390/ma14237168_

Round 1

Reviewer 1 Report

The manuscript is devoted to the electromechanical impedance-based technique of the non-destructive testing (NDT) or evaluation (NDE).

The topic chosen by authors is quite technical and the method aims at industrial applications of searching for the millimeter-size defects. I doubt such a research area fits well into the Materials journal scope. It would look better in a journal like NDT & E International or Structural Health Monitoring.

Nevertheless the research performed is planned, implemented and described well. I have few specific questions to the manuscript:

  1. The figures 6 and 8 have too many windows and may be split into parts.
  2. Fig. 7 deserves an extended discussion on the nature of 140 kHz - 170 kHz features in the impedance curves.

Author Response

The following represent point-by-point answers to the reviewers’ comments. Appropriate revisions are made in the revised manuscript, as explained hereunder. In the revised version of the manuscript, all the revisions are highlighted in yellow.

Reviewer # 01 Comments:

  • The figures 6 and 8 have too many windows and may be split into parts.

Answer: The authors are thankful to the reviewer for the valuable comment. As suggested, in the revised version of the manuscript, the figures 6 and 8 are splitted into parts.

  • 7 deserves an extended discussion on the nature of 140 kHz - 170 kHz features in the impedance curves.

Answer: The authors are thankful to the reviewer for the valuable comment. As suggested, in the revised version of the manuscript, the nature of 140 kHz - 170 kHz features in the impedance curves are discussed in detail.

Finally, the authors wish to thank the reviewer for his constructive remarks, which are well-taken and implemented to improve the clarity and quality of the manuscript. We thank you for the time you put in reviewing our paper and look forward to meet your expectations.

Reviewer 2 Report

On content:

1) Details on weldability of the chosen base metal, steel Q345B are not given. How was its weldability estimated (based on which formula)? It should be stated if the BM is well weldable, conditionally weldable or poorly weldable.

2) Page 4 – Lines 164-165 – there is a statement: "Steel plates must be pre-heated properly before 164 welding to prevent hardened structures and cold cracks". This would point out to the conclusion that this BM is conditionally weldable and that it requires preheating.

Again, which was the preheating temperature and was it determined? Just saying "must be pre-heated properly" is very vague. What does "preheated properly" mean?

3) In table 3 in the last column are given the values of the heat input. How were those values determined?

4) Page 6 – lines 192-193 – there is a statement: "The fatigue behavior of both the specimens was very positive during the entire loading state". What does "very positive fatigue behavior" mean? This is a rather peculiar statement.

What concerns the style of the presentation, there are some problems with English language.

The second point is that the "List of references" is not written according to the Journal's template. Titles of the cited articles contain quotation marks!

The scanned pages with marked errors and suggested corrections are enclosed.

Author Response

The following represent point-by-point answers to the reviewers’ comments. Appropriate revisions are made in the revised manuscript, as explained hereunder. In the revised version of the manuscript, all the revisions are highlighted in green.

Reviewer # 02 Comments:

  • Details on weldability of the chosen base metal, steel Q345B are not given. How was its weldability estimated (based on which formula)? It should be stated if the BM is well weldable, conditionally weldable or poorly weldable.

Answer: The authors are thankful to the reviewer for the valuable comment. As suggested, in the revised version of the manuscript, details of the weldability of the Q345B steel are added in section 3, on pages 4 & 5 of the revised manuscript.

  • Page 4 – Lines 164-165 – there is a statement: "Steel plates must be pre-heated properly before welding to prevent hardened structures and cold cracks". This would point out to the conclusion that this BM is conditionally weldable and that it requires preheating. Again, which was the preheating temperature and was it determined? Just saying "must be pre-heated properly" is very vague. What does "preheated properly" mean?

Answer: The authors are thankful to the reviewer for the valuable comment. As suggested, in the revised version of the manuscript, the pre-heating process is discussed in detail on page 5 of the revised manuscript.

  • In table 3 in the last column are given the values of the heat input. How were those values determined?

Answer: The authors are thankful to the reviewer for the valuable comment. As suggested, in the revised version of the manuscript, the method of calculation of the heat input is discussed on page 5 of the revised manuscript.

  • Page 6 – lines 192-193 – there is a statement: "The fatigue behavior of both the specimens was very positive during the entire loading state". What does "very positive fatigue behavior" mean? This is a rather peculiar statement.

Answer: The authors are thankful to the reviewer for the valuable comment. As suggested, in the revised version of the manuscript, the statement has been corrected.

  • What concerns the style of the presentation, there are some problems with English language.

Answer: The authors are thankful to the reviewer for the valuable comment. As suggested, in the revised version of the manuscript, the problems related to the English language have been corrected.

  • The second point is that the "List of references" is not written according to the Journal's template. Titles of the cited articles contain quotation marks!

Answer: The authors are thankful to the reviewer for the valuable comment. As suggested, in the revised version of the manuscript, the “List of References” is now written in the journal’s format.

  • The scanned pages with marked errors and suggested corrections are enclosed.

Answer: The authors are thankful to the reviewer for the valuable comment. As suggested, in the revised version of the manuscript, the marked errors in the scanned pages are corrected.

Finally, the authors wish to thank the reviewer for his constructive remarks, which are well-taken and implemented to improve the clarity and quality of the manuscript. We thank you for the time you put in reviewing our paper and look forward to meet your expectations.

Reviewer 3 Report

I would like some clarifications on the following points:

  1. The paper presents the mechanical properties and chemical composition of Q345B steel (Tables 1 and 2). However, the source of these data was not given. Are these results of own research, normative data, commercial information, etc.?
  2. The research methodology does not include basic information on the welding method used, welding materials used (normative markings) and welding conditions, ie the value of the preheating temperature. Macroscopic images of welded joints with exposed heat affected zones would also be welcome. Without the above information, in my opinion, it is not possible to carry out a reliable analysis of whether the welding imperfections revealed in the course of the experiments are an inherent feature of the welding processes or are the result of errors made during welding.
  3. The information on the strength tests provided in the manuscript and in Table 4 is very imprecise. There is a lack of basic information regarding the methodology of the conducted tests, i.e. the model of machine used, measuring ranges, the method of determining the value of the force and amplitude used, the number of cycles, methods of controlling the deformation (stress), as well as a specific reference to test standards.
  4. In the manuscript, it correctly refers only to defects (mainly cracks) observed on the surface of the welded joints. In my opinion, there is no discussion regarding the possibility of the applied test method for internal defects in the material under consideration. Adopting such a position would significantly increase the scientific value of the manuscript.

Author Response

The following represent point-by-point answers to the reviewers’ comments. Appropriate revisions are made in the revised manuscript, as explained hereunder. In the revised version of the manuscript, all the revisions are highlighted in Turquoise color.

Reviewer # 03 Comments:

  • The paper presents the mechanical properties and chemical composition of Q345B steel (Tables 1 and 2). However, the source of these data was not given. Are these results of own research, normative data, commercial information, etc.?

Answer: The authors are thankful to the reviewer for the valuable comment. As suggested, in the revised version of the manuscript, the references from which the mechanical properties and chemical composition of Q345B steel were taken are mentioned.

  • The research methodology does not include basic information on the welding method used, welding materials used (normative markings) and welding conditions, ie the value of the preheating temperature. Macroscopic images of welded joints with exposed heat affected zones would also be welcome. Without the above information, in my opinion, it is not possible to carry out a reliable analysis of whether the welding imperfections revealed in the course of the experiments are an inherent feature of the welding processes or are the result of errors made during welding.

Answer: The authors are thankful to the reviewer for the valuable comment. As suggested, in the revised version of the manuscript, information has been added related to the welding method used and pre-heating process on pages 4 & 5. Macroscopic image of the welded joints with exposed heat affected zones is also presented on page 6 of the revised manuscript.

  • The information on the strength tests provided in the manuscript and in Table 4 is very imprecise. There is a lack of basic information regarding the methodology of the conducted tests, i.e. the model of machine used, measuring ranges, the method of determining the value of the force and amplitude used, the number of cycles, methods of controlling the deformation (stress), as well as a specific reference to test standards.

Answer: The authors are thankful to the reviewer for the valuable comment. As suggested, in the revised version of the manuscript, the information on the tests performed in the current study is provided on page 8 of the revised manuscript.

  • In the manuscript, it correctly refers only to defects (mainly cracks) observed on the surface of the welded joints. In my opinion, there is no discussion regarding the possibility of the applied test method for internal defects in the material under consideration. Adopting such a position would significantly increase the scientific value of the manuscript.

Answer: The authors are thankful to the reviewer for the valuable comment. As suggested, in the revised version of the manuscript, a discussion related to the possibility of the internal defects is provided on page 2 of the revised manuscript.

Finally, the authors wish to thank the reviewer for his constructive remarks, which are well-taken and implemented to improve the clarity and quality of the manuscript. We thank you for the time you put in reviewing our paper and look forward to meet your expectations.

Round 2

Reviewer 1 Report

The manuscript was improved by authors. If the topic is appropriate for Materials journal, it can be publishef in the present form.

Author Response

  • The manuscript was improved by authors. If the topic is appropriate for Materials journal, it can be published in the present form.

Answer: The authors are thankful to the reviewer for the valuable comments, which were well-taken and implemented in the revised manuscript and has improved the clarity and quality of the manuscript.

Reviewer 3 Report

Thank you very much for including my comments in the manuscript. Nevertheless, in my opinion, the issue of the macroscopic evaluation of welded joints still needs to be corrected. Thank you for including Fig. 4, however, the image shown on it is not a macroscopic image of the welded joint, but of the entire structural element. On its basis, it is not possible to draw any conclusions regarding the impact of welding operation on the properties of the welded joint. Therefore, I would like to clarify my remark on this point. In my opinion, the manuscript should show a macroscopic image (or microscopic image at low magnification) on the cross-section of an exemplary welded joint. This approach will enable the assessment of the geometrical properties of the welded joint, and additional etching with Adler's reagent (or another) will show the width of the heat-affected zone. It is worth noting, however, that it should not be done in the place marked with the arrow in Fig. 4, because it is the beginning / end (or both of these variants at the same time) of welding, and therefore unrepresentative conditions. In its current form, the information included in the manuscript does not allow for the assessment of the correctness of the welding procedures used, which are independent of the subsequent research procedures.

Author Response

  • Thank you very much for including my comments in the manuscript. Nevertheless, in my opinion, the issue of the macroscopic evaluation of welded joints still needs to be corrected. Thank you for including Fig. 4, however, the image shown on it is not a macroscopic image of the welded joint, but of the entire structural element. On its basis, it is not possible to draw any conclusions regarding the impact of welding operation on the properties of the welded joint. Therefore, I would like to clarify my remark on this point. In my opinion, the manuscript should show a macroscopic image (or microscopic image at low magnification) on the cross-section of an exemplary welded joint. This approach will enable the assessment of the geometrical properties of the welded joint, and additional etching with Adler's reagent (or another) will show the width of the heat-affected zone. It is worth noting, however, that it should not be done in the place marked with the arrow in Fig. 4, because it is the beginning / end (or both of these variants at the same time) of welding, and therefore unrepresentative conditions. In its current form, the information included in the manuscript does not allow for the assessment of the correctness of the welding procedures used, which are independent of the subsequent research procedures.

Answer: The authors are thankful to the reviewer for the valuable comment. As suggested, in the revised version of the manuscript, a macroscopic image of the welded joint is shown in Figure 4 of the revised manuscript and is discussed on pages 5 & 6 of the revised manuscript. As suggested, relevant references to the study area have been discussed in the introduction section on pages 2 & 3 of the revised manuscript. As suggested, the conclusion section has been revised on page 15 of the revised manuscript and is now supported by the results in the revised manuscript.

Finally, the authors wish to thank the reviewer for his constructive remarks, which are well-taken and implemented to improve the clarity and quality of the manuscript. We thank you for the time you put in reviewing our paper and look forward to meet your expectations.
